# In Curative Stereotactic Body Radiation Therapy for Prostate Cancer, There Is a High Possibility That 45 Gy in Five Fractions Will Not Be Tolerated without a Hydrogel Spacer

**DOI:** 10.3390/cancers16081472

**Published:** 2024-04-11

**Authors:** Subaru Sawayanagi, Hideomi Yamashita, Mami Ogita, Taketo Kawai, Yusuke Sato, Haruki Kume

**Affiliations:** 1Department of Radiology, University of Tokyo Hospital, 7-3-1, Hongo, Bunkyo-ku, Tokyo 113-8655, Japan; sawayanagi-tky@umin.ac.jp (S.S.); o-tamami@hotmail.co.jp (M.O.); 2Department of Urology, Graduate School of Medicine, The University of Tokyo, 7-3-1, Hongo, Bunkyo-ku, Tokyo 113-8655, Japan; taketokawai@yahoo.co.jp (T.K.); yusuke_310@infoseek.jp (Y.S.); kumeh-uro@h.u-tokyo.ac.jp (H.K.); 3Department of Urology, School of Medicine, Teikyo University, 2-11-1, Kaga, Itabashi-ku, Tokyo 173-8606, Japan; 4Department of Urology, Tokyo Metropolitan Tama Medical Center, 2-8-29, Musashidai, Fuchu 183-8524, Japan

**Keywords:** prostate cancer, stereotactic body radiation therapy, phase 1 study, dose escalation

## Abstract

**Simple Summary:**

While stereotactic body radiation therapy (SBRT) has become increasingly used for the treatment of non-metastatic prostate cancer since its insurance coverage in Japan in 2016, optimal dose fractionation remains undetermined. This study was a phase 1 dose escalation trial of SBRT, aiming to assess the maximum tolerated dose (MTD) of SBRT using five fractions. Patients were planned to receive SBRT at doses of 42.5, 45, or 47.5 Gy, with toxicity as the primary endpoint. No dose-limiting toxicities were observed at 42.5 Gy, and one patient experienced a grade 4 rectal perforation at 45 Gy, leading to the determination of 42.5 Gy as the MTD. There were no deaths or biochemical recurrences during the follow-up period. This study underscores the need for further trials to ascertain the optimal SBRT dose fractionation, balancing efficacy and safety for non-metastatic prostate cancer treatment.

**Abstract:**

The purpose of this study was to determine the maximum tolerated dose (MTD) for stereotactic body radiation therapy (SBRT) in the treatment of non-metastatic prostate cancer. This study was a phase 1 dose escalation trial conducted in Japan. Patients with histologically proven prostate cancer without lymph nodes or distant metastases were enrolled. The prescribed doses were 42.5, 45, or 47.5 Gy in five fractions. Dose-limiting toxicity (DLT) was defined as grade (G) 3+ gastrointestinal or genitourinary toxicity within 180 days after SBRT completion, and a 6 plus 6 design was used as the method of dose escalation. A total of 16 patients were enrolled, with 6 in the 42.5 Gy group and 10 in the 45 Gy group. No DLT was observed in the 42.5 Gy group. In the 45 Gy group, one patient experienced G3 rectal hemorrhage, and another had G4 rectal perforation, leading to the determination of 42.5 Gy as the MTD. None of the patients experienced biochemical recurrence or death during the follow-up period. We concluded that SBRT for non-metastatic prostate cancer at 42.5 Gy in five fractions could be safely performed, but a total dose of 45 Gy increased severe toxicity.

## 1. Introduction

It is generally known that the α/β ratio of prostate cancer is low [1,2], which has led to hypofractionation in radical external beam radiation therapy (EBRT) for prostate cancer. Moderately hypofractionated EBRT with a single dose of 2.4–4 Gy has been shown to be non-inferior to conventionally fractionated EBRT in several studies [3,4,5,6], making it an alternative option to conventionally fractionated EBRT. Stereotactic body radiation therapy (SBRT) has also shown promising results [7,8] and is one of the curative treatment options for prostate cancer without metastasis. However, long-term results are still limited, and further reports are needed.

In Japan, SBRT has been increasingly used in the treatment of low-, intermediate-, and high-risk prostate cancer since it became covered by insurance in 2016 for localized prostate cancer. Prescription doses of 35–36.25 Gy in five fractions (Frs) over 10 days have been empirically selected in many studies [9,10,11]. According to Version 3. 2024 of Prostate Cancer in the National Comprehensive Cancer Network (NCCN) Guidelines^®^ [12], the preferred dose/Fr for SBRT is 9.5 Gy × 4 Frs, 7.25–8 Gy × 5 Frs, or 6.1 Gy × 7 Frs. Although much experience with SBRT has been accumulated at these dose levels, it has never been established whether this dose is the optimal dose level to be selected for prostate SBRT.

To our knowledge, five phase 1 studies of dose escalation in SBRT for prostate cancer have been reported [13,14,15,16,17]. Four out of them chose five Frs in SBRT [13,14,16,17]. The Harvard University study said that all regimens of 36.25 Gy or 37.5 Gy in five Frs or other regimens were safe and effective for the treatment of low-risk prostate adenocarcinoma [13]. According to the study from Memorial Sloan Kettering Cancer Center [14], all SBRT doses of 32.5 Gy, 35 Gy, 37.5 Gy, or 40 Gy in five Frs were well-tolerated without severe urinary or rectal toxicities. Although Boike et al. completed the dose escalation to 50 Gy as a total dose for SBRT in five Frs without dose-limiting toxicity (DLT) [16], in the subsequent phase 2 trial, they concluded that severe late toxicity increased at total doses greater than 47.5 Gy [18]. Potters et al. concluded that 50 Gy was the maximum tolerated dose (MTD), as there were no DLT [17]. This paper reports the results of a phase 1 dose escalation study of SBRT using five Frs for patients with localized prostate cancer. We started this study with six patients at a prescribed dose of 42.5 Gy in five Frs. Side effects were the primary endpoint of this phase 1 trial, but the incidence of prostate-specific antigen (PSA) recurrence was included as a secondary endpoint. The purpose of this study was to determine what happens to the incidence of acute and late gastrointestinal (GI) and genitourinary (GU) toxicities as the total dose level of SBRT increases.

## 2. Materials and Methods

### 2.1. Patients and Eligibility

This study was part of the institutional review board approved single institution, phase 1 dose escalation trial of SBRT for non-metastatic prostate cancer (UMIN000039444). All patients provided written informed consent before study entry according to the Declaration of Helsinki and national regulations.

Patients with histologically proven prostate cancer without lymph nodes or distant metastases were enrolled in this trial. Histological findings were not required to exclude metastases. Patients had never received any treatment for prostate cancer other than hormonal therapy. The Eastern Cooperative Oncology Group performance status was between 0 and 2. Age was between 20 and 85 years old at the time of informed consent. Patients with a history of inflammatory colitis, pelvic radiotherapy, or rectal surgery were excluded. This study included patients who were clinically unsuitable for hydrogel spacer insertion. In other words, patients with the following conditions: difficulty inserting an ultrasound probe into the rectum; allergy to local anesthetics or midazolam; rectal involvement or dorsal extracapsular invasion of carcinoma; surgical treatment in the lower pelvic region in the past; antithrombotic therapy requiring heparinization during drug interruption. All patients were classified at the time of diagnosis according to the NCCN risk classification [12] using pelvic magnetic resonance imaging (MRI)-based T-stages, biopsy results, and PSA values.

### 2.2. Treatment Planning

All the patients received plain computed tomography (CT) reconstructed with 1 mm slices for treatment planning. The entire pelvic cavity and urethral bulb were included in the imaging area. Patients were placed in the supine position. Patients were encouraged to refrain from urinating for about two hours and to drink water as needed to increase bladder capacity. The goal was to result in a bladder capacity of about 200–250 cc on the planning CT. Laxatives and enemas were administered in advance to reduce rectal volume.

The radiation technique was volumetric modulated arc therapy (VMAT) with energies of 6 MV, and Monaco (Elekta AB, Stockholm, Sweden) was used as the treatment planning system. MRI images acquired under the same conditions were fused to the planning CT, and each organ at risk (OAR) and target volume were outlined. Gross tumor volume (GTV) was defined as a tumor detectable on an MRI. Clinical target volume (CTV) was defined according to the NCCN risk classification. The CTV was the prostate in the low-risk group, the prostate and proximal 1 cm of the seminal vesicles in the intermediate-risk group, the prostate and proximal 2 cm of the seminal vesicles in the high-risk group, and the prostate and the entire seminal vesicles in the case of seminal vesicle invasion. The planning target volume (PTV) was created by extending the CTV by 5 mm in all directions except for 3 mm in the posterior direction. Bilateral femoral heads, bladder, rectum, sigmoid colon, small bowel, penile bulb, and intraprostatic urethra were delineated as organs at risk. The rectum was defined as the portion within 1 cm craniocaudal to the PTV.

The prescribed dose was 42.5, 45, or 47.5 Gy in five Frs, and any dose was prescribed for 95% of the PTV. More than 99% of the PTV had to receive more than 95% of the prescription dose. The bladder wall was limited to 110% of the prescription dose, with no more than 50 cc receiving 20 Gy or greater. The area receiving ≥40 Gy and ≥25 Gy was limited to ≤25% and ≤45% of the rectum, respectively. The femoral heads were not allowed to receive more than 30 Gy. The small bowel was limited to ≤29 Gy, with no more than 10 cc receiving 20 Gy or greater. The maximum intraprostatic urethra dose was limited to ≤105% of the prescription dose. The dose constraints are summarized in Table 1.

### 2.3. Radiotherapy

Before each treatment, the same fixation as the treatment planning CT was performed. Positioning was confirmed by taking a kilovoltage cone beam CT and comparing it with the treatment planning CT. The bladder and rectal capacity were also confirmed to be like those at the time of the treatment planning CT. When the bladder capacity was insufficient, fulfilling the bladder was continued. Radiotherapy was delivered once daily on every other weekday for 5 consecutive days. Any kind of premedication with steroids or α-blockers was not applied.

### 2.4. Study Endpoints and Statistics

This study was designed as a dose-escalation study to determine the MTD of SBRT for localized prostate cancer. DLT was defined as grade (G) 3+ GU or GI toxicity according to the Common Terminology Criteria of Adverse Events version 5.0 within 180 days after SBRT completion. The primary endpoint was the incidence of DLT. Based on the “A plus B design” by Ivanova et al. [19], a 6 plus 6 design was chosen for this study considering that the dose at which the incidence of DLT was 10% or less was appropriate and DLT occurrence of 20% or more was unacceptable. Six patients were assigned to each new dose level. The dose was increased if none of the six patients experienced DLT. When one of them experienced DLT, six more patients were treated at that dose level. Simulations showed that the highest dose with a DLT incidence of 10% or less had the highest probability of being selected as the MTD. The probability that any of the doses with a DLT incidence of 10% or less would be selected as the MTD was greater than 75%. Dose escalation was discontinued if more than one patient experienced DLT at the same dose level or one patient experienced G4+ toxicity at any time in the study period.

Adverse event assessment and PSA measurement were performed before treatment, 2 weeks, 1, 3, 6, 9, and 12 months after the end of treatment, and every 6 months thereafter. Late adverse events were defined as those occurring more than 90 days after treatment, and acute adverse events as those occurring before that. The Kaplan–Meier method was used for overall survival (OS), biochemical recurrence-free survival (bRFS), and cumulative incidence of toxicity analysis. A 2 ng/mL increase in PSA nadir after treatment was defined as a biochemical recurrence [20]. All statistical analyses were performed using R, version 4.3.2.

## 3. Results

A total of 16 patients were enrolled from February 2020 to April 2022. Six patients were assigned to the 42.5 Gy group and 10 to the 45 Gy group. The median follow-up was 39.5 months (range, 17.8–43.7) for the 42.5 Gy group and 30.1 months (16.6–32.7) for the 45 Gy group. Baseline patient characteristics are shown in Table 2. The median age was 72 years (71–83) in the 42.5 Gy group and 71 years (63–84) in the 45 Gy group. Eight patients (50.0%) were classified as intermediate risk and eight (50.0%) as high risk according to the NCCN risk classification. Twelve patients (75.0%) received concurrent androgen deprivation therapy (ADT). No patients experienced biochemical recurrence or died during the observation period.

No DLT was seen among six patients in the 42.5 Gy group. One patient experienced G3 rectal hemorrhage at 5 months after SBRT among the 45 Gy group, requiring the enrollment of 12 patients in the 45 Gy group to determine 45 Gy as the MTD. Since another patient in the 45 Gy group experienced G4 rectal perforation at 13 months after the end of irradiation, dose escalation was discontinued, and 42.5 Gy was determined as the MTD. A total of 10 patients were enrolled in the 45 Gy group and received SBRT until recruitment was discontinued. Except for the two patients mentioned above, no G3+ adverse events were observed during the entire observation period. Acute and late GU and GI toxicities by grade at each dose level are shown in Table 3. Acute G2+ GU toxicity occurred in 33.3% of patients in the 42.5 Gy group and 30.0% in the 45 Gy group. Similarly, acute G2+ GI toxicity occurred in 16.7% and 10.0%, late G2+ GU toxicity occurred in 33.3% and 50.0%, and late G2+ GI toxicity occurred in 33.3% and 70.0%, respectively. The cumulative incidence of G3+ GI toxicity is shown in Figure 1. The 2-year cumulative incidence of G3+ GI toxicity in the 45 Gy group was 20.0% (95% CI 5.4–59.1).

The worst late GI toxicities and characteristics by patient are shown in Table 4, and planned target and OAR doses by patient are shown in Table 5. Patient numbers 9 and 10 experienced G3+ late GI toxicities. Age, diabetes, and PTV did not appear to affect the severity of GI toxicities. While rectal volumes receiving 40 Gy and 25 Gy also did not appear to be related to the severity of toxicity, both were high for the maximum rectal dose, 48.3 Gy and 49.4 Gy, respectively, in the two patients who experienced G3+ late GI toxicities. While no G3+ GU toxicity was observed, patients with G2 GU toxicity were more likely to have a higher maximum bladder and urethral dose.

## 4. Discussion

This study is the first prospective study to investigate the MTD of definitive SBRT in five Frs for non-metastatic prostate cancer in a Japanese population. One G3 and one G4 GI adverse event occurred in the 45 Gy group, respectively, resulting in an MTD of 42.5 Gy in this study. During the short median follow-up period of 36.4 months, none of the 6 patients assigned to the 42.5 Gy group experienced biochemical recurrence or death.

Four phase 1 studies of SBRT in five Frs for prostate cancer have been reported [13,14,16,17]. McBride et al. reported that 36.25 Gy and 37.5 Gy were both safe as total doses in 45 low-risk prostate cancer patients, with G3 late GU toxicity in one patient and G3 late GI toxicity in two patients [13]. In a phase 1 study by Zelefsky et al. in low- and intermediate-risk prostate cancer patients, there were no G3+ acute toxicities, G3 late GU toxicity in 1 of 35 patients in the 40 Gy group, and G3 late GI toxicity in 2 of 36 patients in the 37.5 Gy group [14]. From these results, they concluded that SBRT at total doses of 32.5 to 40 Gy was well tolerated. These two studies are consistent with the results of the present study because they demonstrated the tolerability of SBRT at doses lower than 42.5 Gy, the MTD in the present study. In a multi-institutional study by Hannan et al., G3 and G4 late GU toxicity occurred in 4.9% and 1.6%, respectively, and G3 and G4 late GI toxicity occurred in 6.6% and 3.3%, respectively, of patients receiving a total dose of 50 Gy [18]. As a result, they determined that total doses up to 47.5 Gy were tolerable. The MTD in their study was higher than the 42.5 Gy in the present study, which can be attributed to the difference in treatment methods. Unlike the present study, their patients received dexamethasone before each treatment and α-blockers for 6 weeks from the start of treatment, which may have reduced adverse events. In addition, the intraprostatic marker and intrarectal balloon allowed for a smaller PTV margin, which may have led to dose reductions in risk organs. A phase 1 study by Potters et al. of 26 patients with low- and intermediate-risk prostate cancer successfully increased the total dose up to 50 Gy as the MTD without G3 or higher acute and late toxicity [17]. The MTD in their study was also higher than in the present study. The PTV margins they used were the same as those used in the present study, but the high-dose range for the OARs is expected to be smaller because the CTV did not include the seminal vesicles.

Several randomized controlled trials (RCTs) of conventionally fractionated irradiation for prostate cancer have shown that dose escalation improves bRFS [21,22,23,24,25,26,27]. In a RCT by Pasalic et al., dose escalation from 70 Gy to 78 Gy not only improved biochemical recurrence-free survival but also significantly reduced distant metastases [27]. A decrease in biochemical recurrence with increasing doses has also been reported in SBRT. Zelefsky et al. reported 5-year cumulative biochemical recurrence rates of 15%, 6%, 0%, and 0% for the 32.5 Gy, 35 Gy, 37.5 Gy, and 40 Gy groups, respectively [14]. In a phase 1 study by Hannan et al., the freedom from biochemical failure rate was 90.9%, 100%, and 100% for the 45 Gy, 47.5 Gy, and 50 Gy groups, respectively, at 5 years [18]. In a phase 1 study by Potters et al., there were a total of two cases of biochemical recurrence at 43 and 62 months after SBRT in the 40 Gy group, but not in the 45 Gy and 50 Gy groups; PSA nadir was 0.6, 0.1, and 0.1 ng/mL in the 40 Gy, 45 Gy, and 50 Gy groups, respectively [17]. Based on these reports, dose escalation up to 45–47.5 Gy may improve control of non-metastatic prostate cancer. Although no biochemical recurrence was observed in the present study using a high total dose of SBRT, this may have been due to the shorter follow-up period compared to previous studies.

While the present study used dose escalation for the entire PTV, there is another approach in which dose escalation is given only to the dominant intraprostatic lesion (DIL) identified in imaging studies. A phase III study by Kerkmeijer et al. showed that simultaneous integrated boost (SIB) to the DIL identified on MRI during moderately hypofractionated EBRT for localized prostate cancer improved biochemical disease-free survival but did not significantly increase treatment-related toxicity [28]. Several recent reports on SBRT for prostate cancer have also shown promising results regarding the efficacy and safety of SIB for DIL [29,30,31]. It can theoretically be performed more safely than the overall PTV dose escalation performed in the present study, but there are not many reports on its efficacy, and further studies are expected.

There are various types of dose escalation designs used in phase 1 trials, and the appropriate design must be selected for each trial. Classically, the rule-based design, represented by the 3 plus 3 design, has been widely used in determining the MTD of antitumor drugs [32,33]. Although rule-based designs have been used specifically for cytotoxic antitumor drugs, Araujo et al. found that rule-based designs were also used in 92% of phase 1 studies of molecularly targeted drugs or immunotherapies published between 2014 and 2019 [34]. In general, the accuracy of MTD estimation is higher in model-based designs than in rule-based designs because model-based designs use toxicity information from all patients previously treated with various doses to select the next dose and determine the MTD [35]. However, the model-based designs have the disadvantage of misidentification in dose-response models, and they also have safety concerns since dose levels can be skipped and escalated based on the toxicity results of a single patient. Among the rule-based designs, several modifications of the traditional 3 plus 3 design have been proposed to allow rapid progression of phase 1 trials [36,37], all of which involve the risk of an increase in the number of patients treated at doses higher than the MTD. Due to safety concerns, we used the conventional A plus B design, which requires a prolonged period [19].

There are several limitations in the present study, including the small sample size and short follow-up period since this is a phase 1 study. In this study, DLT was defined as within 180 days of treatment, which does not adequately include the late toxicity that would be most problematic because of dose escalation. As a phase 1 study, it was necessary to limit the timing of DLT. Apart from DLT, we included the occurrence of G4+ toxicity within the entire study period as a condition for discontinuation of dose escalation, but this is not a fair condition because the difference in the follow-up period may affect the incidence rate. In our study, the median follow-up period for the 42.5 Gy group was approximately 10 months longer than for the 45 Gy group, which means that patients in the 42.5 Gy group have a greater opportunity for toxicity to occur. Despite the recent widespread use of hydrogel spacers due to their clinical advantages [38,39,40,41,42,43], the results of this study can only be applied to patients who undergo SBRT without hydrogel spacers. However, there are patients who are not suitable for hydrogel spacer insertion, such as those with extracapsular invasion in the rectal direction, and this study is significant because it helps us to know the MTD for them. The MTD of SBRT with hydrogel spacer for prostate cancer is also being investigated in a separate part of the present study and will be reported.

## 5. Conclusions

We report the results of a phase 1 study of SBRT for non-metastatic prostate cancer at our institution. A total dose of 42.5 Gy in five fractions of SBRT could be safely performed, but a total dose of 45 Gy increased severe toxicity. Additional trials are warranted to determine the optimal dose based on efficacy and safety.

## Figures and Tables

**Figure 1 cancers-16-01472-f001:**
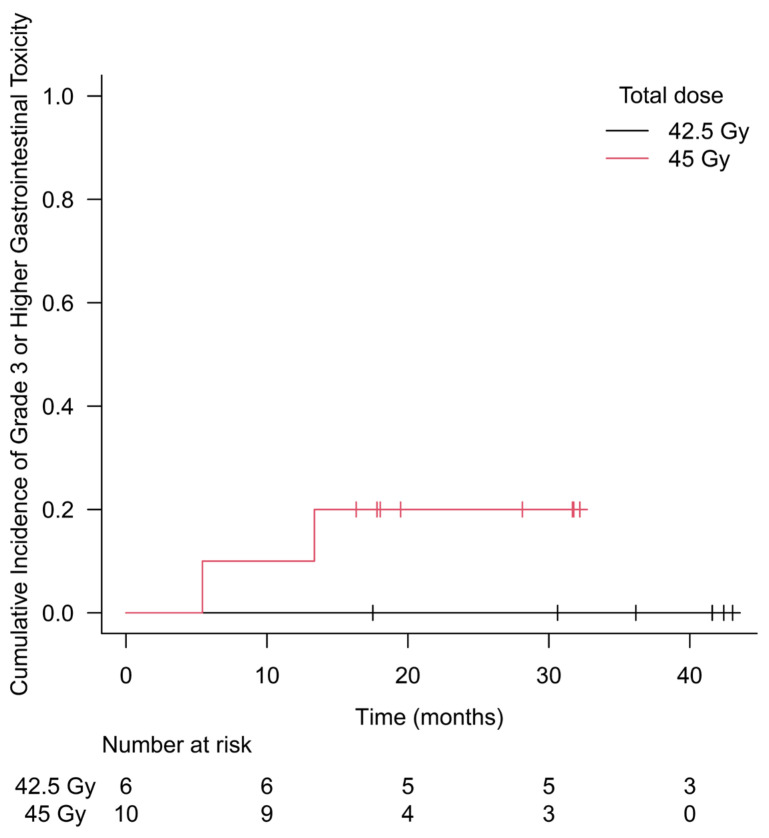
Cumulative incidence of grade 3 or higher gastrointestinal toxicity. No grade 3 or higher toxicity was observed during the follow-up period in the 42.5 Gy group. The 2-year cumulative incidence of grade 3 or higher gastrointestinal toxicity in the 45 Gy group was 20.0% (95% CI 5.4–59.1).

**Table 1 cancers-16-01472-t001:** Dose constraints.

Prescribed Dose, Gy	PTV		Bladder Wall		Rectum ^1^		Femoral Heads	Small Bowel		Urethra
	D95% ^2^, Gy	D99% ^2^, Gy	V20 Gy ^3^, cc	Max dose, Gy	V40 Gy ^3^, %	V25 Gy ^3^, %	Max Dose, Gy	V20 Gy ^3^, cc	Max Dose, Gy	Max Dose, Gy
42.5	42.5	≥40.375	≤50	≤46.75	≤25	≤45	≤30	≤10	≤29	≤44.625
45	45.0	≥42.750	≤50	≤49.50	≤25	≤45	≤30	≤10	≤29	≤47.250
47.5	47.5	≥45.125	≤50	≤52.25	≤25	≤45	≤30	≤10	≤29	≤49.875

PTV: planning target volume. ^1^ The rectum was defined as the portion within 1 cm craniocaudal to the PTV. ^2^ Dx% means the lowest dose of x% receiving the highest dose in the target. ^3^ Vx Gy means the volume of the organ receiving x Gy or more.

**Table 2 cancers-16-01472-t002:** Baseline patient characteristics.

Characteristics	Dose		Total
	42.5 Gy	45 Gy	
Number of patients	6	10	16
Age, years, median (range)	72 (71–83)	71 (63–84)	71.5 (63–84)
Clinical T stage, n (%)			
1c	0 (0%)	0 (0%)	0 (0%)
2a	3 (50.0%)	4 (40.0%)	7 (43.8%)
2b	0 (0%)	0 (0%)	0 (0%)
2c	1 (16.7%)	1 (10.0%)	2 (12.5%)
3a	2 (33.3%)	3 (30.0%)	5 (31.3%)
3b	0 (0%)	2 (20.0%)	2 (12.5%)
4	0 (0%)	0 (0%)	0 (0%)
Grade group, n (%)			
1	0 (0%)	0 (0%)	0 (0%)
2	1 (16.7%)	6 (60.0%)	7 (43.8%)
3	2 (33.3%)	2 (20.0%)	4 (25.0%)
4	0 (0%)	0 (0%)	0 (0%)
5	3 (50.0%)	2 (20.0%)	5 (31.3%)
Initial PSA, ng/mL, median (range)	11.1 (2.85–82.28)	8.89 (3.91–44.24)	9.34 (2.85–82.28)
NCCN risk, n (%)			
Favorable intermediate	1 (16.7%)	1 (10.0%)	2 (12.5%)
Unfavorable intermediate	2 (33.3%)	4 (40.0%)	6 (37.5%)
High	1 (16.7%)	2 (20.0%)	3 (18.8%)
Very high	2 (33.3%)	3 (30.0%)	5 (31.3%)
ADT use, n (%)			
No	2 (33.3%)	2 (20.0%)	4 (25.0%)
Yes	4 (66.7%)	8 (80.0%)	12 (75.0%)

ADT: androgen deprivation therapy; NCCN: National Comprehensive Cancer Network; and PSA: prostate-specific antigen.

**Table 3 cancers-16-01472-t003:** The worst acute and late genitourinary and gastrointestinal toxicities at each dose level.

Highest Grade	42.5 Gy, n (%)				45 Gy, n (%)				Total, n (%)			
	Acute GU	Acute GI	Late GU	Late GI	Acute GU	Acute GI	Late GU	Late GI	Acute GU	Acute GI	Late GU	Late GI
0	0 (0.0%)	3 (50.0%)	0 (0.0%)	2 (33.3%)	2 (20.0%)	2 (20.0%)	3 (30.0%)	2 (20.0%)	2 (12.5%)	5 (51.3%)	3 (18.8%)	4 (25.0%)
1	4 (66.7%)	2 (33.3%)	4 (66.7%)	2 (33.3%)	5 (50.0%)	7 (70.0%)	2 (20.0%)	1 (10.0%)	9 (56.3%)	9 (56.3%)	6 (37.5%)	3 (18.8%)
2	2 (33.3%)	1 (16.7%)	2 (33.3%)	2 (33.3%)	3 (30.0%)	1 (10.0%)	5 (50.0%)	5 (50.0%)	5 (51.3%)	2 (12.5%)	7 (43.8%)	7 (43.8%)
3	0 (0.0%)	0 (0.0%)	0 (0.0%)	0 (0.0%)	0 (0.0%	0 (0.0%)	0 (0.0%)	1 (10.0%)	0 (0.0%)	0 (0.0%)	0 (0.0%)	1 (6.3%)
4	0 (0.0%)	0 (0.0%)	0 (0.0%)	0 (0.0%)	0 (0.0%)	0 (0.0%)	0 (0.0%)	1 (10.0%)	0 (0.0%)	0 (0.0%)	0 (0.0%)	1 (6.3%)

GI―gastrointestinal; GU―genitourinary.

**Table 4 cancers-16-01472-t004:** Worst late GI toxicities and characteristics by patient.

Patient Number	Total Dose, Gy	Highest Grade	Age, Years	Diabetes	PTV, cc	Rectal Volume ^1^, cc
1	42.5	0	74	Yes	78.0	64.0
2	42.5	2	71	Yes	79.3	43.8
3	42.5	1	71	No	61.7	37.2
4	42.5	0	71	No	70.6	42.0
5	42.5	1	83	No	101.2	47.4
6	42.5	2	73	No	62.9	34.2
7	45	0	70	No	124.3	64.4
8	45	2	70	No	85.4	56.8
9	45	4	63	No	81.2	44.4
10	45	3	72	No	133.8	85.7
11	45	2	75	No	66.9	40.8
12	45	2	73	No	90.3	28.8
13	45	1	72	No	69.5	57.5
14	45	0	84	No	77.2	35.6
15	45	2	66	No	102.1	57.6
16	45	2	64	No	71.5	60.0

PTV―planning target volume. ^1^ The rectum was defined as the portion within 1 cm craniocaudal to the PTV.

**Table 5 cancers-16-01472-t005:** Toxicities, planned targets, and OAR doses by patient.

Patient Number	Total Dose, Gy	Highest Grade of GI	Highest Grade of GU	PTV		Bladder Wall		Rectum ^1^		Femoral Heads	Small Bowel		Urethra
				D95% ^2^, Gy	D99% ^2^, Gy	V20 Gy ^3^, cc	Max Dose, Gy	V40 Gy ^3^, %	V25 Gy ^3^, %	Max Dose, Gy	V20 Gy ^3^, cc	Max Dose, Gy	Max Dose, Gy
1	42.5	0	2	42.5	42.1	23.8	45.9	3.0	14.2	23.9	0.0	4.5	44.0
2	42.5	2	1	42.5	41.9	46.6	45.4	8.0	27.9	24.2	0.0	6.6	44.6
3	42.5	1	2	42.5	42.0	36.0	45.9	8.2	23.8	26.1	0.0	2.8	44.6
4	42.5	0	1	42.5	41.8	28.5	45.3	4.2	16.5	25.7	0.0	7.4	44.3
5	42.5	1	1	42.5	41.9	45.6	45.7	14.2	43.1	27.6	0.0	3.6	44.5
6	42.5	2	1	42.5	42.0	48.4	45.7	9.6	32.5	21.0	0.0	6.2	44.1
7	45	0	1	45.0	44.5	42.8	49.0	9.2	27.7	21.8	0.0	12.4	46.6
8	45	2	1	45.0	44.3	38.2	47.9	18.4	40.3	21.4	0.0	8.5	46.6
9	45	4	2	45.0	44.4	44.6	48.8	11.7	30.0	28.7	0.0	5.7	47.2
10	45	3	2	45.0	44.3	47.9	49.1	7.1	22.3	24.0	0.0	7.9	46.7
11	45	2	2	45.0	44.5	49.4	47.9	14.5	35.1	26.9	0.0	9.6	46.9
12	45	2	2	45.0	44.4	42.5	47.6	16.1	35.3	27.0	0.1	24.1	46.6
13	45	1	2	45.0	42.9	18.0	48.6	11.2	23.9	24.2	0.4	27.0	47.1
14	45	0	1	45.0	44.3	37.9	48.3	10.7	26.6	20.9	0.0	7.5	46.9
15	45	2	2	45.0	43.6	26.5	48.4	19.8	42.6	20.9	1.6	28.6	47.2
16	45	2	1	45.0	44.2	34.2	47.9	9.8	21.5	23.8	0.0	9.9	46.9

GI―gastrointestinal; GU―genitourinary; OAR―organ at risk; and PTV―planning target volume. ^1^ The rectum was defined as the portion within 1 cm craniocaudal to the PTV. ^2^ Dx% means the lowest dose of x% receiving the highest dose in the target. ^3^ Vx Gy means the volume of the organ receiving x Gy or more.

## Data Availability

The datasets used and/or analyzed during the current study are available from the corresponding author on reasonable request.

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
