# Peer review of "In Curative Stereotactic Body Radiation Therapy for Prostate Cancer, There Is a High Possibility That 45 Gy in Five Fractions Will Not Be Tolerated without a Hydrogel Spacer"

_cancers, 2024, doi:10.3390/cancers16081472_

Round 1

Reviewer 1 Report

Comments and Suggestions for Authors
I have read with great interest this article concerning the role of dose escalation for prostate SBRT. The authors report the results of a phase I trial enrolling 16 patients. The paper is well written and data are properly collected and presented. However, I have some observations. -How was performed staging for intermediate and high risk prostate cancer? please specify
-Why dorsal extracapsular invasion was considered as an exclusion criterion, while no impact of seminal vesicles invasion? please specify
-The bladder filling protocol seems quite open to heterogeneity, did the authors observe any trouble for bladder volume variations?
-Did the authors apply any kind of premedication with steroids or alpha blockers for the course of SBRT?
I think the Authors should provide a larger background on the role of SBRT in comparison with moderate hypofractionation (also citing the following studies -->
doi: 10.1177/0300891619867846; doi: 10.1007/s40520-019-01243-1.) Moreover, they should also justify the need for entire gland dose-escalation, as several literature studies are reporting the safety and efficacy of boosting only the DIL, with the theoretical advantage of reducing the dose to the entire gland (doi: 10.1016/j.ijrobp.2024.03.009; doi: 10.1007/s00345-024-04876-8; doi: 10.1007/s00066-023-02189-0)

Reviewer 2 Report

Comments and Suggestions for Authors

Both MTD and DLT strongly depend on OAR doses. More detailed information on them "must" be provided. 

Consider providing at least 2 tables:

Table 1 that summarizes the dose prescription and constraints (i.e., lines 107 to 114).

Table 2 that summarizes planned OAR doses & complications for each case.

A minor comment: In line 117, modify "a kilovoltage CT" to "a kilovoltage CBCT" if it was CBCT. Or "a kilovoltage FBCT" in case it was a fan beam for better communication.

Round 2

Reviewer 1 Report

Comments and Suggestions for Authors

I thank the authors for the edits to the manuscript. I have no other observations.